# A Decision-Theoretic Public Health Framework for Heated Tobacco and Nicotine Vaping Products

**DOI:** 10.3390/ijerph192013431

**Published:** 2022-10-18

**Authors:** David T. Levy, Christopher J. Cadham, Yameng Li, Zhe Yuan, Alex C. Liber, Hayoung Oh, Nargiz Travis, Mona Issabakhsh, David T. Sweanor, Luz Maria Sánchez-Romero, Rafael Meza, K. Michael Cummings

**Affiliations:** 1Lombardi Comprehensive Cancer Center, Georgetown University, Washington, DC 20057, USA; 2Department of Health Management and Policy, School of Public Health, University of Michigan, Ann Arbor, MI 48109, USA; 3Centre for Health Law, Policy & Ethics, University of Ottawa, Ottawa, ON K1N 6N5, Canada; 4Faculty of Law, University of Ottawa, Ottawa, ON K1N 6N5, Canada; 5Department of Integrative Oncology, BC Cancer Institute, Vancouver, BC V5Z1L3, Canada; 6Department of Psychiatry and Behavioral Sciences, Medical University of Charleston, Charleston, SC 29425, USA

**Keywords:** heated tobacco products, e-cigarettes, ENDS, decision theory, public health, policy

## Abstract

Markets for nicotine vaping products (NVPs) and heated tobacco products (HTPs) have grown as these products became positioned as harm-reduction alternatives to combusted tobacco products. Herein, we present a public health decision-theoretic framework incorporating different patterns of HTP, NVP, and cigarette use to examine their impacts on population health. Our framework demonstrates that, for individuals who would have otherwise smoked, HTP use may provide public health benefits by enabling cessation or by discouraging smoking initiation and relapse. However, the benefits are reduced if more harmful HTP use replaces less harmful NVP use. HTP use may also negatively impact public health by encouraging smoking by otherwise non-smokers or by encouraging initiation or relapse into smoking. These patterns are directly influenced by industry behavior as well as public policy towards HTPs, NVPs, and cigarettes. While substantial research has been devoted to NVPs, much less is known about HTPs. Better information is needed to more precisely define the health risks of HTPs compared to cigarettes and NVPs, the relative appeal of HTPs to consumers, and the likelihood of later transitioning to smoking or quitting all products. While our analysis provides a framework for gaining that information, it also illustrates the complexities in distinguishing key factors.

## 1. Introduction

The tobacco marketplace has dramatically changed in the last ten years. In the US, while cigarette use has rapidly declined [1], especially among youth [2,3,4], the use of other nicotine delivery products, such as smokeless tobacco and nicotine vaping products (NVPs) [5,6,7,8], has increased. England [9,10,11,12] and Canada [12,13,14] show similar trends in cigarette and NVP use. At the same time, the use of heated tobacco products (HTPs) has rapidly emerged in Japan [15,16,17,18,19,20,21] and the Republic of Korea [22,23,24,25] and is now also becoming more popular in other countries [26,27,28,29,30,31,32]. In many countries, HTPs are readily available along with NVPs [29,31,32]. While sales are currently restricted, young adults [33,34,35] and smokers [36] have previously shown interest in IQOS (an HTP) in the US, suggesting that HTPs may again find a foothold in the market.

With the growth in new products, the nicotine product landscape now contains an array of products that can increase the number of dependent users, but may also act as a substitute for the deadliest of products, cigarettes. A decision-theoretic framework was previously developed to evaluate the public health impacts of NVPs [37,38,39]. With health risks from NVPs less than from smoking [40,41,42,43,44], the framework showed that public health improves when NVPs are used by never smokers who would have instead initiated smoking, by current smokers who would not have otherwise quit smoking, and by former smokers who would have otherwise eventually relapsed. However, public health is worsened when NVP use encourages smoking initiation, discourages smoking cessation, or encourages smoking relapse. That framework [37] has explicitly or implicitly been used in a wide variety of simulation models that have examined the public health implications of smoking relative to NVPs or other potential harm-reducing products [6,38,39,45,46,47,48,49,50,51,52,53].

With the addition of newly emerging nicotine delivery products, such as HTPs, their potential public health implications become considerably more complex. The impact of these new products depends on their use patterns. Because of its similarities with cigarettes (i.e., heating tobacco) [54,55,56] and reduced toxicity [57,58,59,60,61,62,63,64,65,66], HTPs may be a viable substitute for smoking where NVPs have failed. For example, it has been suggested that “Juul (an NVP) appeals to millennials/hipsters and IQOS appeals to slightly older and more affluent smokers.” [67]. However, public health impacts will also depend on the likelihood of transitioning from HTP to cigarette use and HTP health risks relative to both NVPs and cigarettes.

A better understanding of the likely transitions to and from HTP, NVP, and cigarette use is needed to effectively develop tobacco control policies that minimize the harms from nicotine delivery product use. It will be essential to understand how the impact of NVP-, HTP- and cigarette-oriented policies and regulations depends on consumers’ use of the targeted product and other products, and any impact of policies on industry behavior. In particular, HTPs have historically been sold only by cigarette companies, while NVPs are also sold by non-cigarette-producing companies [68,69,70]. Policies may influence industry behavior by cigarette-producing vs. non-cigarette-producing companies.

This paper expands our previous decision-theoretic framework of NVP and cigarette use [37,38,39] to incorporate HTP use as a second potentially harm-reducing product. In presenting the framework, we discuss recent studies addressing the information needed for modeling the public health impact of heated tobacco products. We also consider the role of industry marketing, focusing on cigarette companies, and the role of policies in developing a comprehensive approach to balance the potential harms of cigarette, HTP, and NVP use. Due to the lack of requisite information on transitions between products and relative risks, we do not attempt to model the public impact of HTPs and NVPs. However, the described framework intends to aid in the extension of previous tobacco products simulation models that only consider one potential harm-reducing product. The Discussion section describes the additional information that can be used to help extend previous simulation models.

## 2. General Approach

Because cigarette use continues to be the nicotine delivery product most harmful to public health [71,72], our decision-theoretic framework focuses on the potential impact of NVPs and HTPs transitions to and from cigarette use (smoking). Since health outcomes depend primarily on long-term regular use [64], public health impacts are based on regular use, such as use maintained for at least one year [73]. Transitions to regular NVP and HTP use may involve shifts back and forth between experimental HTP and NVP use, but, for simplicity, we focus on transitions to either regular HTP or regular NVP use and not dual-use of HTPs and NVPs. Although NVP and HTP health risks are uncertain, our analysis considers their risk relative to cigarettes. Evidence indicates that NVP use [74,75,76,77] and HTP use [57,58,59,60,61,62,63,64,65,66] are likely much less harmful than cigarette use. NVP use is often estimated at 5% to 15% of the excess mortality risks of cigarettes [6,40,41,42,43,44,78], although there is considerable controversy on the precise level of the difference [44,79,80]. HTPs are likely more harmful than NVPs, with some estimates ranging from 1.5 to 2 times more harmful than NVPs [43,61,65,81,82,83], implying HTP risks at about 7.5% to 30% of the excess mortality risks of cigarettes. Due to difficulties in identifying regular patterns of multi-product use [84], we do not distinguish dual (with either NVPs or HTPs) from exclusive cigarette use and assume the same health risks for dual-use and exclusive smoking. We assume that the health risks of exclusive smoking/dual use are the highest, followed by those of exclusive HTP and then NVP use.

We also extend our previous framework to consider industry behavior. Cigarette markets are an oligopoly in most countries, providing a highly profitable product [68]. In contrast, NVP markets are generally more competitive with many firms. These markets generally include non-cigarette companies with strong business incentives to target cigarette smokers as well as cigarette companies diversifying their products [69,70]. The HTP market typically contains few firms (typically just cigarette firms) [17,27,28,85,86,87,88,89,90,91,92] and is subject to entry barriers (e.g., proprietary technology [69,70]), and thus firms are likely to face minimal competition and receive greater profits from selling HTPs than NVPs.

To reduce the complexity of the analysis and focus on two potential harm reduction products that have already gained acceptance in many countries, we do not consider oral nicotine pouches [93,94,95,96] or other emerging products. However, discussed below, many of the same issues would arise when considering these other products.

## 3. Decision-Theoretic Framework

We separately consider initiation from never users, cessation from current smokers, and relapse from former smokers, as shown in Figure 1, Figure 2 and Figure 3. The relative impact of final smoking, NVP, and HTP outcomes on public health (e.g., the change in smoking-, NVP- and HTP-attributable deaths) is indicated by shades of green to distinguish potential health gains and shades of red to distinguish potential health losses, with darker shades indicating a greater mortality risk. In each of the analyses, we first consider whether individuals initially engage in regular NVP or HTP use. We then consider potential future transitions (e.g., after one year of NVP/HTP use) from regular NVP or HTP use to exclusive/dual cigarette use, remaining NVP or HTP users, or no use.

### 3.1. Never Users

As in our previous framework [37,38,39], the public health impact of NVP and HTP use by those beginning as never smokers depends on the counterfactual scenarios of whether or not these individuals would have otherwise initiated smoking if NVPs and HTPs were not available. The top branch of Figure 1 at node 1.1 is the counterfactual where never users would have otherwise initiated smoking. There are four potential paths of HTP and NVP use. First, in the absence of HTP or NVP use, the individual simply becomes a smoker (node 1.11), but public health is unaffected since the would-be smoker remains a smoker. Second, the use of HTPs by otherwise smokers (node 1.12) may yield public gains if the never smoker was not inclined to try or had tried but discontinued NVP use (e.g., due to dissatisfaction). In this instance, public health gains derive from replacing smoking initiation with HTP use when NVPs would not have been used. If otherwise smokers would have instead used NVPs (node 1.13), public health impacts depend on whether NVPs are in fact used. Third, if otherwise smokers who would have used NVPs instead use HTPs (node 1.131), e.g., due to industry marketing or public policies, HTP use may still yield public health benefits relative to cigarette use. However, due to greater health risk from HTPs than NVPs, less public health benefit is derived than if NVPs were used (node 1.132), the fourth alternative. Thus, public health gains from HTP use are greatest when NVP use would not have otherwise replaced smoking.

The second main branch of Figure 1 at node 1.2 denotes the counterfactual where never smokers would not have otherwise initiated smoking. NVP and HTP use yields public health losses, because their use expands the pool of nicotine delivery product users. Lack of HTP or NVP use implies no product use, and, consequently, no smoking or harm (node 1.21). Reflecting the increased health risk of HTP compared to NVP use, HTP use (nodes 1.22 and 1.231) shows greater public health losses than NVP use (node 1.232). The greatest loss occurs when HTPs are used even though NVPs would have been preferred in the absence of industry marketing or public policies (node 1.231). Later transitions to exclusive/dual cigarette use may be more likely for HTPs than NVPs due to their similarity to cigarettes, i.e., HTPs heat tobacco [56] whereas NVPs contain primarily nicotine vapor [96]. While direct evidence on whether HTPs act as a gateway to smoking is limited [96] and youth HTP users are less likely than NVP users to quit cigarettes [25]. Although some studies indicate NVP use may precede smoking [96], population-level studies find that declines in youth and young adult smoking have rapidly accelerated with greater NVP use [3,4,97,98,99]. Those initiating HTP use may also be more likely than NVP initiators to continue their use rather than quit all product use, as indicated by higher levels of dependence on HTPs [100,101] compared to NVPs [102,103,104]. While evidence to date is limited, initial HTP use appears more likely than initial NVP use to lead to exclusive/dual cigarette use and less likely to lead to no use. As such, Figure 1 shows that best final outcomes tend to occur as a result of initial NVP use followed by initial HTP use.

In the above analysis, we considered initial transitions to exclusive regular HTP or NVP use. The public health impact of these choices will ultimately depend on any further transitions in later years to cigarette smoking or quitting NVP or HTP use (as shown in the terminal nodes of Figure 1). Regular use of either product may lead to one of three terminal nodes: (1) exclusive or dual cigarette use, (2) exclusive (continued) NVP or HTP use or (3) non-use of any product. Later transitions to exclusive/dual cigarette use may be more likely for HTPs than NVPs due to their similarity to cigarettes, i.e., HTPs heat tobacco [56] whereas NVPs heat a nicotine-containing liquid [56]. While direct evidence on whether HTPs act as a gateway to smoking is limited, studies find that youth HTP use often precedes smokings [25,97]. Although some studies indicate that NVP use may precede smoking [96], population-level studies find that declines in youth and young adult smoking have rapidly accelerated with greater NVP use since the emergence of NVPs [1,3,4,98,99], suggesting that NVP use is not affecting or may even be enhancing declines in youth cigarette smoking. Those initiating HTP use may also be more likely than NVP initiators to continue their use rather than quit all product use, as indicated by higher levels of dependence on HTPs [100,101] compared to NVPs [102,103,104]. While evidence to date is limited, initial HTP use appears more likely than initial NVP use to lead to exclusive/dual cigarette use and less likely to lead to no use. As shown in Figure 1, better final outcomes tend to occur with initial NVP use.

Cigarette firms may directly influence future transitions of never smokers. They have incentives to promote HTPs over NVPs due to: (1) higher potential relative profitability of HTPs, (2) a greater likelihood that HTP users will transition to their most profitable product, cigarettes, and (3) a greater likelihood of continuing rather than quitting HTP compared to NVP use. As suggested by the 4-Ps (Product, Promotion, Price, Place) framework [105], cigarette companies may:(1) develop HTPs particularly desirable to youth and young adults, e.g., via specific flavors. (2) target advertising to youth and young adults, (3) charge lower prices to youth and young adults, e.g., through discounting, and (4) target stores and internet locations most frequented by youth and young adults.

### 3.2. Current Smokers

The public health impact of NVPs and HTPs on current cigarette smokers depends on the counterfactual scenario of whether or not smokers would have quit in the absence of HTPs or NVPs. Node 2.1 in Figure 2 denotes the counterfactual when smokers would not have otherwise quit, i.e., become former smokers. The public health impact is neutral at node 2.1 because the smoker maintains cigarette use without NVPs or HTPs. If current smokers would not have otherwise used NVPs, HTPs yield public health benefits by providing smokers an additional alternative to help them quit cigarette use (node 2.12), e.g., by providing better nicotine delivery [50] or other desired consumer attributes (e.g., taste) compared to NVPs. When smokers would have otherwise used NVPs to quit smoking (node 2.13), HTP use still yields public health gains (node 2.131), but the gains are less than from NVP use (node 2.132) since increased health risk of HTPs offsets some of the gains compared to NVP use.

Node 2.2 in Figure 2 denotes public health losses under the counterfactual where smokers would have otherwise quit smoking in the absence of HTPs and NVPs. The potential public health losses are generally less if NVPs are used (node 2.232). HTPs could yield greater losses (node 2.22), especially if NVPs were preferred to HTPs (node 2.231).

As shown in Figure 2, the ultimate public health impact of HTP and NVP use depends on future transitions to final nodes of dual or exclusive smoker, continued exclusive NVP or HTP use, or no use. Studies indicate that HTP use is associated with reduced intent to quit [106] and successful smoking cessation [65,96,107,108,109,110], while NVPs are associated with smoking quit success [110,111]. The nascent literature finds that HTPs are often used with cigarettes [18,19,21,22,30,112,113], suggesting that HTP use may be more likely than NVPs to lead to exclusive smoking or dual-use. HTP users [100,109] also appear less likely than NVP users [102,103,104] to eventually quit their use. Thus, HTP use may ultimately lead to worse public health outcomes than NVPs due to less likelihood of remaining a former smoker, increased likelihood of continued smoking, and less likelihood of quitting all products (i.e., no cigarette, NVP, or HTP use).

Cigarettes are inexpensive to produce and generate high-profit margins relative to other consumer products [68]. Thus, cigarette companies have a strong incentive to protect their profits from cigarette sales that might be lost from customers switching to HTPs or NVPs. In particular, they may encourage dual use by smokers [17,27,28,84,85,86,87,88,89,90,91,92], e.g., especially when indoor smoking is restricted [114,115]. When faced with the likely loss of cigarette customers to NVPs, cigarette companies will likely encourage HTP over NVP use. As in our analysis of never smokers, cigarette companies may: (1) modify product characteristics (nicotine content or flavoring), (2) increase promotions to target specific populations (e.g., those of low socioeconomic status [SES] or mental health issues), (3) discount prices, and (4) influence product placement. In particular, cigarette companies may target potential NVP users by advertising HTPs as a better substitute for smoking than NVPs [56], focusing especially on their cigarette customers [28]. Cigarette companies are also well-positioned to target smokers in mass-market retail using shelf-space contracts with retailers [68,69] and online through their websites or discussion groups [70,116].

### 3.3. Former Smokers

The impact of NVP and HTP use on former smokers will depend on the counterfactual of whether former cigarette smokers would have (Figure 3, node 3.1) or would not have (Figure 3, node 3.2) otherwise relapsed in the absence of HTPs or NVPs. When former smokers would have otherwise relapsed, the greatest public health benefit tends to occur with NVP use (node 3.132), followed by HTP use when NVPs would not have otherwise been used (node 3.12), and finally when HTPs, although less desirable, are used instead of NVPs (node 3.131). When former smokers would not have otherwise relapsed, losses to public health are generally least with NVP use (node 3.232), followed by HTP when preferred (node 3.22) and finally HTP used instead of NVPs (nodes 3.231).

In terms of later transitions to smoking or exclusive NVP or HTP use, and remaining former smokers, some evidence indicates that HTP users [96] may be more likely than NVP users [117] to relapse to smoking. As with current smokers, cigarette companies are well-situated to target former smokers to increase the likelihood of HTP over NVP use and encourage relapse.

### 3.4. Summary

The potential health gains from HTP use are generally greatest when used by otherwise smokers or continuing smokers to move away from smoking and where NVP use would not have prevented smoking. Similarly, public health losses accrue when HTP use does not prevent smoking, especially if NVPs would not have otherwise been used. The overall public health gains and losses depend on the size of each smoking status group and the likelihood of HTP or NVP users later transitioning to cigarette use or no product use. Each of these transitions will also depend on industry marketing as described above and how consumers and industry respond to public policies towards NVPs, HTPs, and cigarettes as described below.

## 4. Regulatory Framework

Policy evaluations tend to focus on the impact of a particular policy on the use of the targeted tobacco product and do not consider their impact on other products [118,119,120,121,122]. In developing an overall nicotine product strategy consistent with public health goals, the severity of policies targeting each product should be proportionate to their overall risks [123]. To accomplish that aim, a better understanding of relative product risks is needed [124]. However, as suggested by our framework above, overall risks depend not only on each product’s health risk, but also on HTP and NVP availability and consumer preferences, and potential transitions from HTPs and NVPs to cigarette or no nicotine product use. A higher likelihood of transition from HTP use than from NVP use to smoking may justify stricter policies for HTPs than NVPs, e.g., a higher cigarette or HTP tax relative to NVP taxes.

Policies that reduce the appeal of specific products may induce consumers to switch to the use of other products. Stronger NVP-oriented policies relative to HTP-oriented policies may cause those who would have otherwise used NVPs to instead use HTPs. For example, regulations limiting particular NVP flavors (e.g., US Food and Drug Administration [FDA] disapproval of Pre-market Tobacco Product Applications [PMTAs] for menthol or mint NVPs while allowing menthol or mint HTPs) [125] may encourage HTP use when NVPs would have otherwise been preferred. Our analysis also suggests the importance of perceived risks of HTPs relative to cigarettes and NVPs. Government policies, such as through government websites, media campaigns, and health warning requirements, may also directly influence the perception of risks [126,127,128,129] both in terms of consumers using HTPs or NVPs instead of cigarettes and in terms of using HTPs instead of NVPs.

While valuations of tobacco control policies generally focus on their immediate effects on consumers, stricter policies towards HTP than NVPs may also be justified to offset the effect of industry marketing to promote HTP over NVP use. Stricter NVP relative to HTP policies may also have indirect effects through their impact on the viability of firms. Regulations that weaken non-cigarette firms and reduce competition may provide cigarette companies greater control of the overall nicotine delivery product market [69,70]. For example, policies restricting NVP flavors without similar restrictions on HTPs may reduce NVP use and ultimately limit the ability of non-cigarette firms to introduce new products or even survive. With reduced competition from non-cigarette firms, cigarette companies may be better positioned to encourage smoking initiation, discourage smoking cessation, and encourage smokers to switch to HTPs rather than NVPs.

Our analysis is in terms of public health impacts, but policymakers may have other goals, such as the elimination of cigarette companies or cigarette use or even more broadly the use of all nicotine delivery products. Such “regulatory stances” [130] classify a regulatory framework by its intent to change the size of a given market in the future compared to the present. For example, based on the evidence of substantially higher health risks from cigarettes than either NVPs or HTPs, a regulatory stance towards cigarettes that is contractionist (reducing the share of that market in the economy) or even prohibitionist (intent to reduce the market’s size to zero) may be warranted. Placing the cigarette market at a large enough competitive disadvantage compared to the newer markets could provide much of the market momentum towards these new products achieving policy goals. To the extent that HTPs are used where NVPs had not been effective and do not later lead to cigarette use, an expansionist (increasing the size of the market) or permissive (setting no market size goal) stance towards HTPs may be warranted in the short-run, while moves towards contraction or even prohibition could be considered over the longer run if HTP markets are later found to be unacceptably harmful. Focusing on market size and its connection to public health may help prioritize those policies that most efficaciously improve public health.

## 5. Discussion

Our decision-theoretic framework shows how HTP use can result in public health gains under certain scenarios, but the impact depends on a complex set of factors, including relative health risks, use patterns, industry behavior, and public policy. If HTPs are used by those who would not have otherwise used NVPs, they may provide public health benefits by enabling additional smokers to quit or discourage smoking initiation. However, public health benefits are generally reduced if HTP use replaces less harmful NVP use, and especially if that use encourages concurrent or exclusive cigarette use by never smokers who would not have otherwise initiated smoking, smokers who would have otherwise quit, or those who would not have otherwise relapsed. The ultimate impact also depends on the likelihood of future transitions to cigarette use or to quitting all nicotine product use.

Our analysis provides a framework for further empirical analysis and modeling. Currently, harm reduction models [6,39,45,46,47,48,49,50,51,52,53] focus on one potential harm-reducing product. In particular, the two extant HTP simulation models [48,49], both supported by industry, only include HTPs and cigarettes. We have summarized the information from previous studies, but our review suggests the need for further information required to develop models that capture the complexity when more than one harm-reducing product is available.

Separate studies have considered the users of HTP and NVPs among never, current and former smokers, but studies have not generally considered the HTP and NVP use in the context of a setting with multiple harm-reducing products nor the relationship between HTP and NVP users (i.e., overlapping characteristics of users). As illustrated by our counterfactual analysis, the public health impact of harm reduction products will also depend on incorporating risk factors that distinguish would-be smokers among never smokers, would-be quitters among current smokers, and would-be relapsers among former smokers. Studies indicate that these products’ use patterns depend on a complex array of attitudes towards risks and the options available to users [97,131,132,133]. Further, limited attention has been given to later transitions from regular HTP use to cigarette use or no use (i.e., cessation from HTPs and no further cigarette use).

Limited attention has been given in the previous literature to appropriate measures of HTP use, especially in relation to NVP use. We have distinguished initial regular HTP and NVP use from later transitions. The appropriate definitions regarding the duration and intensity of use should depend on empirical analyses and a definition that fits the requirements for gauging public health impacts. Transitions from initial HTP and NVP use to later cigarette use or no product use will similarly require empirical analysis for determining the appropriate time frame.

In our analysis above, we assumed that NVPs pose less health risk than HTPs and that HTP risks are lower than cigarette risks. Over time, as the long-term health consequences of both NVPs and HTPs are better understood, the relative risks may change, and thus the public health implications of HTP vs. NVP transitions may change. While, for simplicity, we did not distinguish dual cigarette and NVP or HTP use from exclusive cigarette use, such analyses should also consider the importance of dual use.

The public health impact of HTP use will depend on how industry behavior impacts the initiation and cessation of NVPs, HTPs and cigarettes. In particular, the profitability of HTPs and cigarettes also plays a role in our analyses of industry behavior, particularly in terms of industry marketing. HTP markets vary widely from country to country. In the US, HTPs were previously sold by Altria, but their IQOS sales were halted due to a patent dispute. Recently, Philip Morris International (PMI) obtained FDA approval to market IQOS 3.0 and may be poised to enter the US market [33,34,35]. Unlike Altria, PMI does not sell cigarettes in the US, which may lead to a stronger stance towards HTP use. In other countries, cigarette companies will likely have a greater incentive to protect the use of their most profitable products, currently cigarettes, although profit margins on HTPs also appear to be relatively high [134].

Use patterns will also depend on public policies towards HTPs, as well as public policies towards other products. Their direct impact on use patterns, such as through taxes, enforcement of minimum purchase age, and flavor restrictions, and their indirect impacts, such as through influencing market structure, merit particular consideration. With more stringent policies towards NVPs than HTPs, cigarette companies may face less competition from non-cigarette companies selling NVPs, and thus increased marketing of HTPs and cigarettes. The impact of NVP-oriented and HTP-oriented policies on industry structure and competition between cigarette-producing and non-cigarette NVP firms should be considered in developing a regulatory framework.

Finally, we have not considered modern oral nicotine pouches [93,94,95,96] or other emerging nicotine delivery products. Their public health impact, like that of HTPs, would depend on their risks and use patterns relative to cigarette, NVP and HTP use, as well as industry behavior and government policy. A recent paper [135] provides a decision-theoretic framework for oral nicotine products (snus) similar to the framework for evaluating the public health impact of NVP relative to cigarette use [37,38,39], but still focuses on two products. With the availability of more than two potentially harm-reducing products, public health benefits will tend to increase if such product use replaces additional never smokers who would have initiated smoking, current smokers who would not have quit, or former smokers who would have relapsed. Nevertheless, the complexity increases when evaluating the impact of each additional product relative to that of cigarettes and other potentially harm-reducing products. Consequently, it may be pragmatic to aggregate harm reducing-products in attempting to model the overall public health impact of multiple harm-reducing products. While the complexity of public health analyses increases with each additional product considered, the potential impact of emerging products merits attention, particularly in terms of their substitutability with HTPs and NVPs.

## 6. Conclusions

To date, substantial research has been devoted to NVPs. Much less is known about HTPs. Information is needed to more precisely define their health risks relative to other products, their appeal to consumers relative to NVPs and cigarettes, and the likelihood of transitioning from HTPs to cigarette use and to quitting all products. While our framework provides a structure for these analyses, it also shows the difficulty in disentangling those relationships. The potential substitution between NVPs and HTPs and their propensity to encourage smoking and quit all product use is particularly complex. Finally, our analysis highlights the importance of incorporating rigorous analyses of industry behavior and the impact of policies on use and industry behavior. Accurate information on each of these factors will be needed to develop comprehensive and effective strategies to promote public health in the increasingly complex nicotine product landscape.

## Figures and Tables

**Figure 1 ijerph-19-13431-f001:**
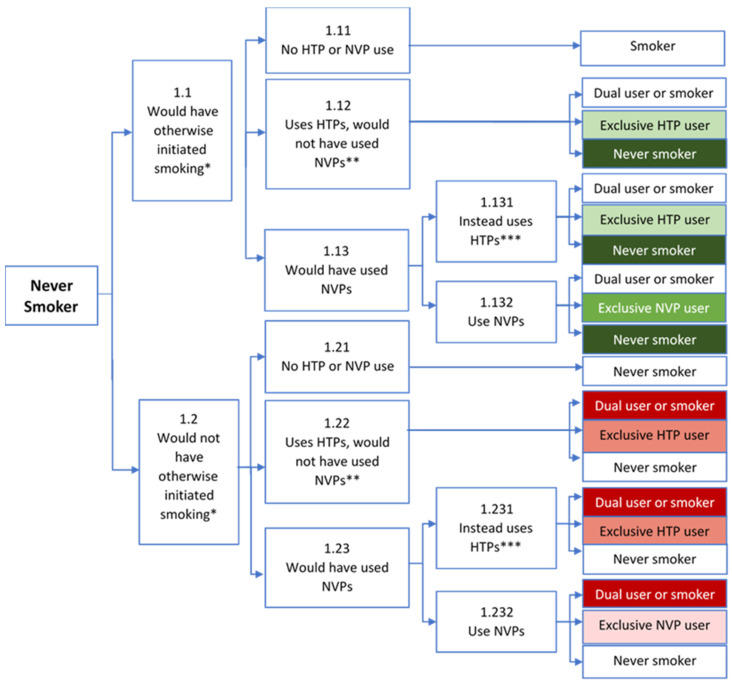
Public Health Impact of NVP and HTP Use Among Former Smokers. Notes: HTP = Heated Tobacco Product, NVP = Nicotine Vaping Product. * Based on the counterfactual of whether smoking initiation would have occurred in the absence of both HTPs and NVPs. ** HTPs are a more desirable alternative to NVPs, e.g., tried NVPs and not found desirable or satisfactory. *** Would have otherwise preferred NVPs, but diverted from use due to industry behavior or government policy. **Green** indicates public health benefit; **Red** indicates public health loss. Darker shades indicate a greater impact.

**Figure 2 ijerph-19-13431-f002:**
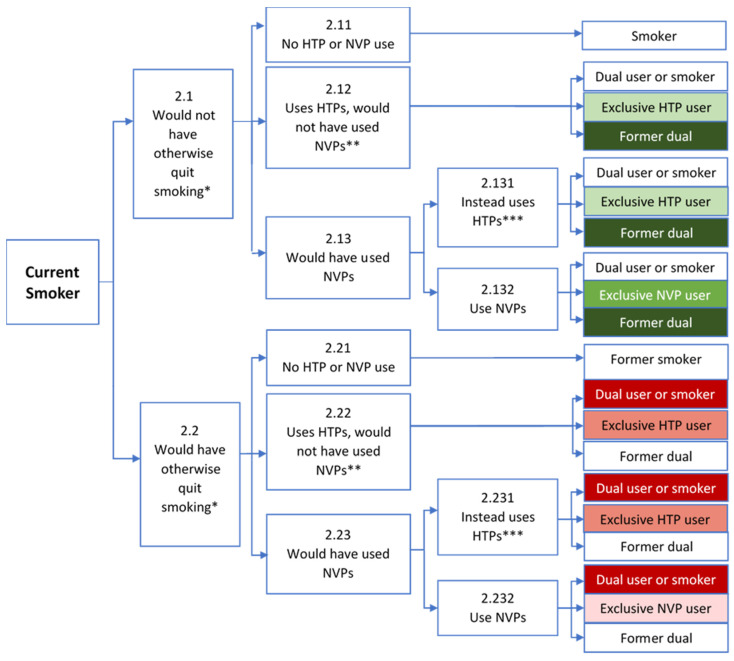
Public Health Impact of NVP and HTP Use Among Current Smokers. Notes: HTP = Heated Tobacco Product, NVP = Nicotine Vaping Product. * Based on the counterfactual of whether smoking cessation would have occurred in the absence of both HTPs and NVPs. ** HTPs are a more desirable alternative to NVPs, e.g., tried NVPs and not found desirable or satisfactory. *** Would have otherwise preferred NVPs, but diverted from use due to industry behavior or government policy. **Green** indicates public health benefit; **Red** indicates public health loss. Darker shades indicate a greater impact.

**Figure 3 ijerph-19-13431-f003:**
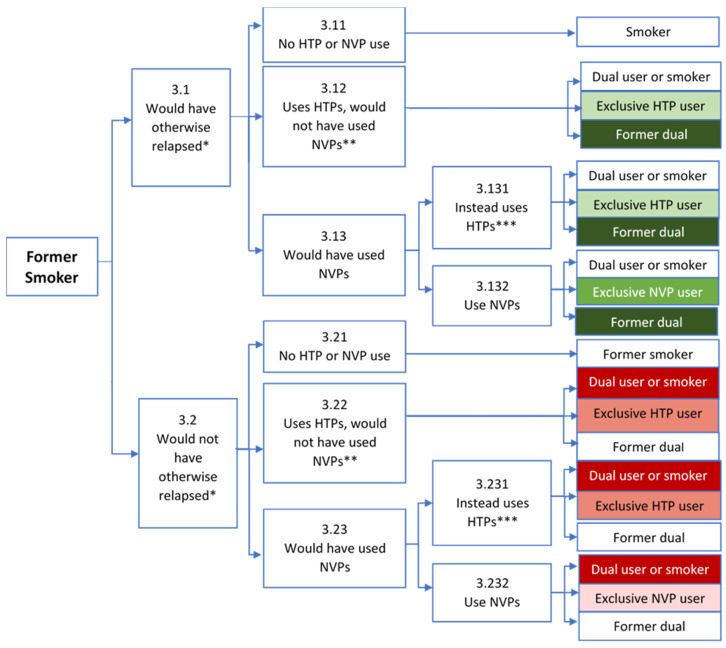
Public Health Impact of NVP and HTP Use Among Former Smokers. Notes: HTP = Heated Tobacco Product, NVP = Nicotine Vaping Product. * Based on the counterfactual of whether smoking relapse would have occurred in the absence of both HTPs and NVPs. ** HTPs are a more desirable alternative to NVPs, e.g., tried NVPs and not found desirable or satisfactory. *** Would have otherwise preferred NVPs, but diverted from use due to industry behavior or government policy. **Green** indicates public health benefit; **Red** indicates public health loss. Darker shades indicate a greater impact.

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
