# Peer review of "A Decision-Theoretic Public Health Framework for Heated Tobacco and Nicotine Vaping Products"

_ijerph, 2022, doi:10.3390/ijerph192013431_

Round 1
Reviewer 1 Report
Review of “A Decision-Theoretic Public Health Framework For Heated Tobacco and Nicotine Vaping Products“
This article presents and discusses a decision-theoretical framework for assessing the public health consequences of adding nicotine vaping products (NVPs) and heated tobacco products (HTPs) to the tobacco market.
The authors build upon their earlier works on frameworks for evaluating the public health impact of e-cigarettes and other vaporized nicotine products, and expand this framework to include HTPs.
I believe this kind of balancing of the potential outcomes of market changes is necessary, and that the addition of HTP is important and worth publishing.
I do not have any major issues with the manuscript, but some minor.
- It could be noted that the the authors may be too careful when discussing the relative risks between cigarettes, HTPs and NVPs, for example on page 2: “Evidence indicates that NVP use 65-68 and HTP use 48-85 56 are likely less harmful than cigarette use, but that HTPs are likely more harmful than 86 NVPs. 43,54,56,69-71”. This is correct, but camouflages the large differences, especially between cigarette smoking and NVPs. The main consequence may be that some readers may question the importance of such a framework if the products sound relatively similar with regards to health risks. Maybe the authors could include a discussion of the number of never users of any product who would need take up HTPs or NVPs to offset the public health gain of 1 smoker who switched to using HTPs or NVPs, or a similar example?
- Although not really a criticism of the present work, future evaluations of pros and cons of new nicotine/tobacco products will need to incorporate an increasing number of products. The complexity will therefor increase dramatically. One example is nicotine pouches, which may be the next addition to NVPs and HTPs, and which has already been discussed separately in a similar type of paper (Lund and Vedøy 2021). One question the authors could have discussed is how we should deal with such increasingly complex frameworks, and if it is possible to come up with a qualitatively different approach.
- The text in node 1.21 in Figure 1 is not readable.
4. Lund, Karl Erik and Tord Finne Vedøy. 2021. "A Conceptual Framework for Assessing the Public Health Effects from Snus and Novel Non-Combustible Nicotine Products." Nordic Studies on Alcohol and Drugs:14550725211021248. doi: 10.1177/14550725211021248.
Author Response
A Decision-Theoretic Public Health Framework for Heated Tobacco and
Nicotine Vaping Products (Manuscript ID: ijerph-1905031)
We thank the reviewer for the valuable comments and suggestions. We have highlighted changes in the manuscript using track changes for easier identification. Responses to the reviewers’ comments below are identified with “Response.” Any changes to the manuscript are marked here in italics.
Response to Reviewer 1 Comments
This article presents and discusses a decision-theoretical framework for assessing the public health consequences of adding nicotine vaping products (NVPs) and heated tobacco products (HTPs) to the tobacco market. The authors build upon their earlier works on frameworks for evaluating the public health impact of e-cigarettes and other vaporized nicotine products and expand this framework to include HTPs. I believe this kind of balancing of the potential outcomes of market changes is necessary and that the addition of HTP is important and worth publishing. I do not have any major issues with the manuscript, but some minor.
Point 1: It could be noted that the authors may be too careful when discussing the relative risks between cigarettes, HTPs and NVPs, for example on page 2: "Evidence indicates that NVP use 65-68 and HTP use 48-85 56 are likely less harmful than cigarette use, but that HTPs are likely more harmful than 86 NVPs. 43,54,56,69-71". This is correct, but camouflages the large differences, especially between cigarette smoking and NVPs. The main consequence may be that some readers may question the importance of such a framework if the products sound relatively similar with regards to health risks. Maybe the authors could include a discussion of the number of never users of any product who would need to take up HTPs or NVPs to offset the public health gain of 1 smoker who switched to using HTPs or NVPs, or a similar example?
Response 1: We now describe the likely ranges of magnitudes of NVP and HTP risks relative to cigarettes, but also mention the controversy over these values. We added the following text to the General Approach section on page 5: Evidence indicates that NVP use74-77and HTP use57-65 are likely much less harmful than cigarette use. NVP use is often estimated at 5% to 15% of the excess mortality risks of cigarettes,6,40-44,78 although there is considerable controversy on the precise level of the difference.44,79,80 HTPs are likely more harmful than NVPs, with some estimates ranging from 1.5 to 2 times more harmful than NVPs,37-39,42,43,63,65,81-83 implying HTP risks at about 7.5% to 30% of the excess mortality risks of cigarettes.
We do not include a discussion of the number of never users of any product who would need to take up HTPs or NVPs to offset the public health gain of a smoker switching to using HTPs or NVPs, because: 1) those who take up NVPs or HTPs at earlier stages may later smoke cigarettes, 2) the risks of a smoker quitting cigarettes depend on the number of years quit (i.e., a person who switches to NVPS above age 40 will still have risks similar to that of smokers for the first five years), and 3) more generally relative risks are dependent on age, so that increased NVP use at younger ages is not comparable to smokers switching to cigarettes at older ages. For these reasons, we expect that a comparison of youth initiation to adult switching would be inaccurate and potentially misleading.
Point 2: Although not really a criticism of the present work, future evaluations of the pros and cons of new nicotine/tobacco products will need to incorporate an increasing number of products. The complexity will therefore increase dramatically. One example is nicotine pouches, which may be the next addition to NVPs and HTPs, and which has already been discussed separately in a similar type of paper (Lund and Vedøy 2021). One question the authors could have discussed is how we should deal with such increasingly complex frameworks, and if it is possible to come up with a qualitatively different approach.
Response 2: We added a reference to the Lund and Vedoy (2021) paper and elaborated on our discussion of the complexity of adding new products. In the second last paragraph of the Discussion section, we added: A recent paper137 provides a decision-theoretic framework for oral nicotine products (snus) similar to the framework for evaluating the public health impact of NVP relative to cigarette use,37-39 but still focuses on two products. With the availability of more than two potentially harm-reducing products, public health benefits will tend to increase if such product use replaces additional never smokers who would have initiated smoking, current smokers who would not have quit, or former smokers who would have relapsed. Nevertheless, the complexity increases when evaluating the impact of each additional product relative to that of cigarettes and other potentially harm-reducing products. Consequently, it may be pragmatic to aggregate harm reducing-products in attempting to model the overall public health impact of multiple harm-reducing products.
Point 3: The text in node 1.21 in Figure 1 is not readable.
Response 3: We thank the reviewer for bringing this to our attention. The box was compressed in the conversion to IJERPH format. We expanded the text box to make the text readable.
Lund, Karl Erik and Tord Finne Vedøy. 2021. "A Conceptual Framework for Assessing the Public Health Effects from Snus and Novel Non-Combustible Nicotine Products." Nordic Studies on Alcohol and Drugs:14550725211021248. DOI: 10.1177/14550725211021248.

Reviewer 2 Report
The paper presents a decision theoretic framework for modeling public health consequences of competition between conventional cigarettes, heated tobacco products, and nicotine aerosol products like electronic cigarettes. Three models are presented – one each for non-smokers, smokers, and ex-smokers – and discussed with regard to counterfactual branches and nodes. The conclusions are that the structures are “a framework for further empirical analysis and modeling,” and that it is possible for competing products to result in public health gains, mitigated or enhanced by policy and industrial decisions regarding the products. Extensive references are offered, especially for various claims that might provoke critics of harm reduction.
I am extremely sympathetic to the authors’ perspective.
The three models are nearly identical, and the discussion of the slight differences in outcomes based on the starting population is long. I think the whole presentation could be more succinct. Perhaps the purpose could be served better by presenting a single version of the model without specifying the starting population and without the color coded terminal nodes, and adding a table with population on one axis (e.g. columns) and path and node ID on the second axis (rows), then color coding the cells.
The paper suggests no probabilities, costs, or utilities in spite of the extensive references. The authors probably want to cite this paper from a multitude of other papers that will add estimates of those data to reach conclusions. Nevertheless, it would be useful for other analysts if the authors could suggest a base case in this paper. The authors may object that this is beyond the scope of the current manuscript, but there have been a multitude of reports that managed to present models attempting to explore very similar topics – including analyses of HTP – and then run them. The manuscript does not cite these efforts or place the current model in that broader context.
Without a base case of updated data or historical context, it is a little hard to understand what this paper adds. The main idea that safer products will expand use and encourage relapse while still benefiting public health has been in the public forum for years, even if it is not widely accepted.
Here is a short list of references that seem relevant but that I did not find in the manuscript.
Ahmad & Billimek, DOI 10.1111/j.1539-6924.2005.00647.x
Feirman, et al, DOI 10.1093/ntr/ntv104
Lee, et al, DOI 10.1093/ntr/ntaa102
Sumner, DOI 10.1136/tc.12.2.124
Author Response
A Decision-Theoretic Public Health Framework for Heated Tobacco and
Nicotine Vaping Products (Manuscript ID: ijerph-1905031)
We thank the reviewer for the valuable comments and suggestions. We have highlighted changes in the manuscript using track changes for easier identification. Responses to the reviewers’ comments below are identified with “Response.” Any changes to the manuscript are marked here in italics.
Response to Reviewer 2 comments
The paper presents a decision-theoretic framework for modeling public health consequences of competition between conventional cigarettes, heated tobacco products, and nicotine aerosol products like electronic cigarettes. Three models are presented – one each for non-smokers, smokers, and ex-smokers – and discussed with regard to counterfactual branches and nodes. The conclusions are that the structures are "a framework for further empirical analysis and simulation modeling," and that it is possible for competing products to result in public health gains, mitigated or enhanced by policy and industrial decisions regarding the products. Extensive references are offered, especially for various claims that might provoke critics of harm reduction.
I am extremely sympathetic to the author's perspective.
Point 1: The three models are nearly identical, and the discussion of the slight differences in outcomes based on the starting population is long. I think the whole presentation could be more succinct. Perhaps the purpose could be served better by presenting a single version of the model without specifying the starting population and without the color coded terminal nodes, and adding a table with population on one axis (e.g. columns) and path and node ID on the second axis (rows), then color coding the cells.
Response 1: We retained the different figures, because of important differences between the transitions from never, current and former smokers. In particular, the initial node setting out the counterfactual differs for never, current and former smokers, and final outcomes also differ (e.g., former smokers in the smokers and former smoker figures vs. never smokers in the never smoker figures). Moreover, the diagrams follow a similar structure and color coding as earlier publications of decision frameworks, so keeping them consistent facilitates comparisons across studies. However, following the reviewer’s suggestion, we have now shortened the discussion for each of the individual frameworks in the decision-theoretic framework section. We also note that the summary at the end of that section provides a succinct discussion of the overlapping concepts for the three frameworks.
Point 2: The paper suggests no probabilities, costs, or utilities in spite of the extensive references. The authors probably want to cite this paper from a multitude of other papers that will add estimates of those data to reach conclusions. Nevertheless, it would be useful for other analysts if the authors could suggest a base case in this paper. The authors may object that this is beyond the scope of the current manuscript, but there have been a multitude of reports that managed to present models attempting to explore very similar topics – including analyses of HTP – and then run them. The manuscript does not cite these efforts or place the current model in that broader context.
Response 2: An essential part of this paper is the finding that the requisite information to develop a credible model is quite limited. Further, even a very tentative model would require a substantial increase in the length of the paper, and would, in our view, be very complicated. Therefore, we do not provide a specific base case analysis. We elaborate in response to point 3 below the changes that we have made to make the purpose of the paper clearer.
Point 3:Without a base case of updated data or historical context, it is a little hard to understand what this paper adds. The main idea that safer products will expand use and encourage relapse while still benefiting public health has been in the public forum for years, even if it is not widely accepted.
Response 3: Our paper goes beyond previous analyses by providing a framework to describe the increased complexity involved in evaluating the impact on public health of adding a potentially harm-reducing product (HTPs) to an existing potentially harm-reducing product (NVPs). In order to make the purpose of this framework clearer to the reader, we added the following sentence at the end of the second paragraph of the Introduction to describe our previous (NVP) framework paper, "That framework has explicitly or implicitly been used in a wide variety of simulation models that have examined the public health implications of smoking relative to NVPs or other potential harm-reducing products.6,38,39,45-53" In order to make the purpose of this paper clearer to the reader, we expanded the last paragraph of the Introduction to state:
This paper expands our previous decision-theoretic framework of NVP and cigarette use37-39 to incorporate HTP use as a second potentially harm-reducing product. In presenting the framework, we discuss recent studies addressing the information needed for modeling the public health impact of heated tobacco products. We also consider the role of industry marketing, focusing on cigarette companies, and the role of policies in developing a comprehensive approach to balance the potential harms of cigarette, HTP, and NVP use. Due to the lack of requisite information on transitions between products and relative risks, we do not attempt to model the public impact of HTPs and NVPs. However, the described framework intends to aid in the extension of previous tobacco products simulation models that only consider one potential harm-reducing product. The Discussion section describes the additional information that can be used to help extend previous simulation models.
The Discussion section has also been revised to emphasize the additional information that can aid in the extension of previous simulation models that only consider one potential harm-reducing product. In particular, we have expanded the previous discussion to the following three paragraphs (second, third and fourth paragraph of the Discussion section):
Our analysis provides a framework for further empirical analysis and modeling. Currently, harm reduction models6,39,45-53 focus on one potential harm-reducing product. In particular, the two extant HTP simulation models,48,49 both supported by industry, only include HTPs and cigarettes. We have summarized the information from previous studies, but our review suggests the need for further information required to develop models that capture the complexity when more than one harm-reducing product is available.
Separate studies have considered the users of HTP and NVPs among never, current and former smokers, studies have not generally considered HTP and NVP use in the context of a setting with multiple harm-reducing products nor the relationship between HTP and NVP users (i.e., overlapping characteristics of users). As illustrated by our counterfactual analysis, the public health impact of harm reduction products will also depend on incorporating risk factors that distinguish would-be smokers among never smokers, would-be quitters among current smokers, and would-be relapsers among former smokers. Studies indicate that these products' use patterns depend on a complex array of attitudes towards risks and the options available to users.97,133-135 Further, limited attention has been given to later transitions from regular HTP use to cigarette use or no use (i.e., cessation from HTPs and no further cigarette use).
Limited attention has been given in the previous literature regarding the appropriate measures of HTP use, especially in relationship to NVP use. We have distinguished initial regular HTP and NVP use from later transitions. The appropriate definitions regarding the duration and intensity of use should depend on empirical analyses and a definition that fits the requirements for gauging public health impacts. Transitions from initial HTP and NVP use to later cigarette use or no product use will similarly require empirical analysis for determining the appropriate time frame.
In our analysis above, we assumed that NVPs pose less health risk than HTPs and that HTP risks are lower than cigarette risks. Over time, as the long-term health consequences of both NVPs and HTPs are better understood, the relative risks may change, and thus the public health implications of HTP vs. NVP transitions may change. While, for simplicity, we did not distinguish dual cigarette and NVP or HTP use from exclusive cigarette use, such analyses should also consider the importance of dual use.
Point 4:Here is a short list of references that seem relevant but that I did not find in the manuscript.
Ahmad & Billimek, DOI 10.1111/j.1539-6924.2005.00647.x
Feirman, et al, DOI 10.1093/ntr/ntv104
Lee, et al, DOI 10.1093/ntr/ntaa102
Sumner, DOI 10.1136/tc.12.2.124
Response 4: We added the suggested citations (except Sumner, et al, which consider nicotine inhalers) along with many other citations to the last sentence of the second paragraph of the Introduction on page 3: That framework has been implicitly or explicitly used in a wide variety of models that have examined the public health implications of smoking relative to NVPs or other potential harm-reducing products.6,39,45-53